# Genetic Risk and Phenotype Correlation of Primary Open-Angle Glaucoma Based on Rho-Kinase Gene Polymorphisms

**DOI:** 10.3390/jcm10091953

**Published:** 2021-05-01

**Authors:** Yong-Woo Kim, Eunoo Bak, Seoyoung Wy, Seung-Chan Lee, Yu-Jeong Kim, Young-Kook Kim, Ki-Ho Park, Jin-Wook Jeoung

**Affiliations:** Department of Ophthalmology, Seoul National University Hospital, Seoul 03080, Korea; yongwookim@snu.ac.kr (Y.-W.K.); eunoo629@gmail.com (E.B.); wsy445@gmail.com (S.W.); tmdcks87@naver.com (S.-C.L.); yjkimhappy@hanmail.net (Y.-J.K.); md092@naver.com (Y.-K.K.); kihopark@snu.ac.kr (K.-H.P.)

**Keywords:** rho-associated coiled-coil kinase (ROCK), single nucleotide polymorphism, primary open-angle glaucoma

## Abstract

Rho-associated coiled-coil kinase (ROCK) signaling can affect glaucoma risk by regulating trabecular meshwork outflow. We investigated the effect of *ROCK* gene polymorphism on the risks of primary open-angle glaucoma (POAG) and POAG-related phenotypes including intraocular pressure (IOP) in a Korean population. A total of 24 single-nucleotide polymorphisms (SNPs) from *ROCK1* and *ROCK2* were selected and genotyped for 363 POAG patients and 213 healthy controls. Among the 363 POAG patients, 282 were normal-tension glaucoma (NTG, baseline IOP ≤ 21 mmHg) and 81 were high-tension glaucoma (HTG, baseline IOP > 21 mmHg). The SNPs rs288979, rs1006881, rs35996865, rs10083915, and rs11873284 in *ROCK1* (tagged to each other, *r*^2^ = 1) were nominally associated with risk of HTG (OR = 0.52, *p* = 0.045). However, there were no SNPs that were significantly associated with the risk of NTG. In the genotype-phenotype correlation analysis, the SNPs rs2230773 and rs3771106 in *ROCK2* were significantly correlated with central corneal thickness (CCT)-adjusted IOP (*p* = 0.024) and axial length (AXL; *p* = 0.024), respectively. The present data implicated the role of *ROCK* in POAG development, and as such, can serve as a good reference for upcoming Rho/ROCK-pathway-related studies on POAG.

## 1. Introduction

Rho-associated coiled-coil kinase (*ROCK*) is a member of the serine/threonine protein kinase family that serves as a major downstream effector of the Rho pathway [1,2]. ROCK attaches to Rho and forms a Rho/ROCK complex that regulates actin–myosin dynamics and is involved in multiple physiological functions, such as cell contraction, migration, proliferation, angiogenesis, chemotaxis, neural protection, and vasodilatation [3,4]. In humans, ROCK has two isoforms, ROCK1 and ROCK2, each of which is separately encoded on chromosome 18 (18q11.1) and chromosome 2 (2p24), respectively [5].

ROCK molecules are ubiquitous in all cellular tissues and organs, including ocular tissues such as the cornea, trabecular meshwork (TM), iris, and retina [6]. Recently, ROCK has drawn increased clinical interest in the field of glaucoma due to the improved understanding of its role in intraocular pressure (IOP) regulation [7]. ROCK inhibitors can lower IOP by modifying the cytoskeleton in Schlemm’s canal and relaxing smooth muscle cells in the TM. It is well documented that TM cells express both ROCK1 and ROCK2 as well as downstream effectors of the ROCK pathway, including myosin light chain (MLC), Lin-11/Isl-1/Mec-3 (LIM) kinase, and cofilin [8]. These findings support the hypothesis that ROCK inhibitors can lower IOP by reducing outflow resistance. ROCK inhibitors are also known for their neuroprotective effect on retinal ganglion cells (RGC) via improved ocular blood flow, RGC survival, and axonal regeneration [9,10]. Therefore, ROCK inhibitors currently are under investigation as potential therapeutic targets for primary open-angle glaucoma (POAG).

In this regard, the *ROCK* gene polymorphisms may play a role in stratifying the risk of POAG on a genetic basis. The purpose of the present study, accordingly, was to investigate possible associations between *ROCK* gene variants and POAG development in a Korean population. In addition, the genotype–phenotype correlation between *ROCK* gene variants and clinical features of POAG, including IOP, were explored.

## 2. Methods

The present study was undertaken as a part of the GLAU-GENDISK (GLAUcoma GENe DIscovery Study in Korea) project, which is an ongoing prospective study designed and inaugurated in 2011 [11]. The primary objective of the GLAU-GENDISK project is to investigate and identify novel genetic variants for various types of glaucoma in a Korean population. The secondary objectives include the establishment of the genotype–phenotype relationships in glaucoma patients and the construction of new disease prediction models. This study was approved by the Seoul National University Hospital Institutional Review Board and followed the tenets of the Declaration of Helsinki (1964). Written informed consent was obtained from each of the enrolled subjects. 

### 2.1. Study Subjects

All of the subjects included in this analysis were Korean. They included 363 patients with POAG and 213 healthy controls, all 576 of whom had been enrolled in the GLAU-GENDISK. POAG was defined as the presence of glaucomatous optic disc changes with corresponding glaucomatous visual field (VF) defects and an open-angle confirmed by gonioscopy. Glaucomatous optic disc changes were defined as neuroretinal rim thinning, notching, excavation, or retinal nerve fiber layer (RNFL) defects. Glaucomatous VF defects were defined as (1) glaucoma hemifield test values outside the normal limits, (2) three or more abnormal points with a probability of being normal of *p* < 0.05, of which at least one point has a pattern deviation of *p* < 0.01, or (3) a pattern standard deviation of *p* < 0.05. The VF defects were confirmed on two consecutive reliable tests (fixation loss rate ≤ 20%, false-positive and false-negative error rates ≤ 25%). The baseline IOP value was defined as the mean of at least two measurements before initiation of IOP-lowering treatment. Based on the baseline IOP values, high-tension glaucoma (HTG) eyes were defined as POAG eyes with a baseline IOP of greater than 21 mm Hg, and normal-tension glaucoma (NTG) eyes were defined as POAG eyes with a baseline IOP of less than or equal to 21 mm Hg.

The present study excluded subjects with (1) a history of retinal diseases such as age-related macular degeneration, epiretinal membrane, or diabetic retinopathy; or (2) insufficient measurement of baseline IOP. 

The POAG patients in the GLAU-GENDISK cohort underwent a complete ophthalmic examination, including a visual acuity assessment, slit-lamp biomicroscopy, gonioscopy, Goldmann applanation tonometry, refractions, dilated fundus examination, disc stereophotography, red-free fundus photography using a digital fundus camera (VISUCAM, Carl Zeiss Meditec, Inc., Dublin, CA, USA), and standard automated perimetry (Humphrey C 24-2 SITA-Standard visual field; Carl Zeiss Meditec, Inc.). The CCT (Pocket II; Quantel Medical, Clermont-Ferrand, France) and AXL (AXIS-II Ultrasonic Biometer; Quantel Medical S.A., Bozeman, MT, USA) were measured as well. A 200 × 200 optic disc cube scan and a 200 × 200 macular cube scan were performed using Cirrus HD-OCT (Carl-Zeiss Meditec, Inc., Dublin, CA, USA), and the average peripapillary RNFL and macular ganglion cell-inner plexiform layer (GCIPL) thicknesses were measured with the built-in analysis algorithm (software version 6.0; Carl Zeiss Meditec, Inc., Dublin, CA, USA). For the POAG patients, the eye with the worse visual field mean deviation (VF MD) was selected for the analysis. For the healthy control, one eye was randomly selected using the sample function in R (R version 3.6.1., available at: http://www.r-project.org; accessed on 20 March 2020).

### 2.2. Target SNP and Genotyping

Candidate single-nucleotide polymorphisms (SNPs) of *ROCK1* and *ROCK2* were selected from reported SNPs that had been previously investigated for association with various diseases in other ethnicities [12]. A total of 24 SNPs (12 from *ROCK1* and another 12 from *ROCK2*) were selected and genotyped (Table 1). Genotyping reactions were performed using the BioMark HD system (Fluidigm 192.24 SNPtypeTM, San Francisco, CA, USA). The primer pools were designed for specific target amplifications as well as allele-specific and locus-specific primers for detection of candidate SNPs (Appendix A). The additional workflow was conducted according to the manufacturer’s instructions for use of the Integrated IFC Controller RX, FC1 Cycler, and EP1 Reader (Fluidigm Corp., San Francisco, CA, USA). Signal intensities for genotype calling were scanned using the EP1 data collection and SNP Genotyping analysis software (version 4, Fluidigm Corp., San Francisco, CA, USA).

### 2.3. Data Analysis

The SNP genotype frequencies were examined for Hardy–Weinberg equilibrium based on the corresponding chi-squared statistics. Continuous variables were compared between the groups by Student’s *t*-test. Data were analyzed using unconditional logistic regression, controlling for age and sex, in order to calculate the odds ratio (OR) as an estimate of the relative risk of POAG associated with the SNP genotype. The key POAG-related phenotypes derived from the 363 eyes of the 363 POAG patients, including IOP, history of disc hemorrhage, mean deviation (MD) of VF, AXL, refraction, rim area, disc area, average cup-to-disc ratio (C/D), vertical C/D, cup volume, average RNFL thickness, average GCIPL thickness, and family history of glaucoma, were evaluated for SNP-genotype correlation after adjustment for age and sex (additive model). In the case of IOP, the additive model was built with adjustment for CCT as well as age and sex. For average RNFL and GCIPL thicknesses, signal strength was also considered for the additive model. Statistical analysis was conducted by R software (R version 3.6.1., available at: http://www.r-project.org; accessed 20 March 2020).

## 3. Results

The present study included 363 POAG patients and 213 healthy controls. There were no significant differences in age (54.0 ± 13.7 vs. 54.6 ± 9.7 years, *p* = 0.55, values are mean ± standard deviation) or sex (female, 180 (49.6%) vs. 93 (43.7%), *p* = 0.17) between the two groups. Among the 363 POAG patients, 282 were NTG and 81 were HTG. The baseline IOP was 15.3 ± 3.0 mm Hg and 26.0 ± 6.5 mm Hg, respectively. The HTG eyes had significantly thicker CCT (546.9 ± 30.2 µM) than did the NTG eyes (532.9 ± 35.9 µm, *p* < 0.001). The HTG eyes also had thinner average RNFL (64.3 ± 14.3 µm) and GCIPL (63.5 ± 11.0 µM) thicknesses and lower MD of VF (−14.5 ± 10.0 dB) (all *p*-s < 0.05). The NTG eyes had a higher number of histories of disc hemorrhage (36 (12.8%)) than did the HTG eyes (3 (3.7%), *p* = 0.034). A Comparison of the ophthalmic demographics between NTG and HTG is provided in Table 2.

All of the SNPs were in Hardy–Weinberg equilibrium (all *p*-s > 0.05). The SNPs rs34945852, rs35768389, rs190769228, and rs1130157 from *ROCK2* were monomorphic in the present population, and therefore, minor alleles were not identified. The following SNPs were found to be in linkage disequilibrium (LD) (*r*^2^ > 0.95): *ROCK1*, SNPs rs75122528, rs1481280, rs2847081, rs8089184, rs288979, rs1006881, rs35996865, rs10083915, and rs11873284; *ROCK2*, SNPs rs965665, rs10178332, and rs6755196.

In the association analysis, none of the SNPs from *ROCK1/ROCK2* were significantly associated with risk of POAG (Table 3). In the subgroup analysis, the SNPs rs288979, rs1006881, rs35996865, rs10083915, and rs11873284 in *ROCK1* (tagged to each other, *r*^2^ = 1) were nominally associated with risk of HTG (OR = 0.52, *p* = 0.045). However, there were no SNPs that were significantly associated with risk of NTG (Table 4).

In the genotype-phenotype correlation analysis, the SNP rs2230773 in *ROCK2* was significantly correlated with CCT-adjusted IOP (*p* = 0.024). The average IOP for the major homozygote (CC) was 17.4 ± 5.8 mmHg, and for the heterozygotes (CT), 20.7 ± 8.6 mmHg (Figure 1). The minor homozygote (TT) was not found. The SNP rs3771106 in *ROCK2* was significantly correlated with AXL (*p* = 0.024). The average AXL for the major homozygote (GG) was 24.41 ± 1.45 mm, for the heterozygotes (GA) 24.81 ± 1.58 mm, and for the minor homozygotes (AA), 24.90 ± 1.91 mm (Figure 1). None of the other POAG-related phenotypes showed any significant correlations with the target SNP genotypes.

In the genotype-phenotype correlation analysis, the SNP rs2230773 in *ROCK2* was significantly correlated with CCT-adjusted IOP (*p* = 0.024). The average IOP for the major homozygote (CC) was 17.4 ± 5.8 mm Hg, and for the heterozygotes (CT), 20.7 ± 8.6 mm Hg. The minor homozygote (TT) was not found. The SNP rs3771106 in *ROCK2* was significantly correlated with AXL (*p* = 0.024). The average AXL for the major homozygote (GG) was 24.41 ± 1.45 mm, for the heterozygotes (GA) 24.81 ± 1.58 mm, and for the minor homozygotes (AA), 24.90 ± 1.91 mm. 

## 4. Discussion

The present study investigated the effect of *ROCK* gene polymorphism on the risk of POAG (also NTG and HTG, respectively), and correlated that risk with the relevant clinical factors in a Korean population. Although the data failed to find significant SNPs associated with POAG risk, a subgroup analysis based on baseline IOP revealed that some SNPs in *ROCK1* (rs288979, rs1006881, rs35996865, rs10083915, and rs11873284) may be associated with higher-baseline-IOP POAG (i.e., HTG). 

ROCK inhibitors are known to reduce IOP by altering TM outflow resistance with the relaxation of smooth muscle cells [6,7]. Besides, ROCK inhibitors are beneficial to glaucoma patients with effects that are independent of IOP: it improves the blood flow to the optic nerve head (via relaxation of vascular endothelial smooth muscle cells) and promotes axonal regeneration [3,9]. To date, two ROCK inhibitors have been approved for clinical use: ripasudil in Japan and netarsudil in the United States. Despite growing clinical interest in ROCK inhibitors as novel therapeutic targets for glaucoma, investigations into *ROCK* gene polymorphisms for POAG risk have been few. Demiryürek et al. [13] for the first time investigated the effect of *ROCK* gene polymorphism on POAG risk. They genotyped 8 SNPs in *ROCK1* and *ROCK2* from 179 POAG patients and 182 healthy controls, but failed to find any significant associations with POAG risk. Our group expanded the number of target SNPs (from 8 to 24) and genotyped for a larger population. We showed that the SNPs rs288979, rs1006881, rs35996865, rs10083915, and rs11873284 in *ROCK1* (tagged to each other, *r*^2^ = 1) were nominally associated with risk of HTG. These SNPs have been reported to be associated with other systemic diseases as well: tetralogy of Fallot for rs288979 [14]; ischemic stroke for rs1006881 and rs10083915 [15], and metabolic syndrome [16], clear cell renal cell carcinoma [17] or systemic sclerosis [18] for rs35996865. As these SNPs were found to be tagged to each other (*r*^2^ = 1) in the present population, they may be in LD with a causative variant rather than being directly causative of POAG risk. Future full sequencing of *ROCK1* or *ROCK2* will further validate the POAG risk variants.

The absence of significant associations of *ROCK* gene variants with POAG risk may imply that other effectors in the Rho/ROCK pathway have a role in POAG development. Springelkamp et al. [19] demonstrated that the SNP rs58073046, located within the gene *ARHGEF12*, was significantly associated with IOP as well as risk of POAG (especially HTG). ARHGEF12 (Rho guanine nucleotide exchange factor (GEF) 12) regulates RhoA GTPases to activate ROCK function, thereby affecting IOP and POAG risk. As clinical evidence for IOP control in POAG patients accumulates, further epigenetics and metabolomics studies promise to uncover complex gene and protein interactions in the Rho/ROCK pathway. Our findings can serve as good reference data for these upcoming studies.

In the present genotype–phenotype correlation analysis, the SNPs rs2230773 and rs3771106 in *ROCK2* were significantly correlated with CCT-adjusted IOP and AXL, respectively. The clinical significance of these two SNPs has not yet been established. As the SNP rs2230773 is known to be a synonymous variant, another causal variant near it may have a role in IOP balancing. The SNP rs3771106 is an intronic variant that is known to be tagged with SNPs rs2230774, rs1515219, rs726843, and rs978906. This also suggests that a causal variant may exist in LD with this SNP.

The present study has the following limitations. First, only a small SNP fraction relative to the size of the *ROCK* gene was selected and genotyped for *ROCK* polymorphism. This may have affected the negative association with POAG risk. Further inclusion of SNPs or full sequencing analysis could better elucidate the causal variants for POAG. Second, a relatively small number of HTG patients compared to NTG patients may have biased the present association results. Due to the low prevalence of HTG in Korea, the majority of POAG patients had their baseline IOP lower than 21 mm Hg. Therefore, caution must be taken to interpret current findings from HTG patients, and further study of larger HTG population is needed. Lastly, all of the subjects included in this analysis were Korean, and so our results cannot be generalized to other ethnicities. East Asian countries including Korea have high proportions of NTG among POAG patients, and thus, subjects from these populations may have different glaucoma characteristics from those of subjects representative of other races or regions [20].

## 5. Conclusions

In conclusion, *ROCK1*-related variants were nominally associated with risk of HTG, but not with NTG, for a Korean population. Also, SNPs rs2230773 and rs3771106 in *ROCK2* showed significant correlation with CCT-adjusted IOP and AXL, respectively. The present data supports the role of *ROCK* in POAG pathogenesis and the relevant key clinical phenotypes, including IOP. This study will serve as an important reference for further investigations of the efficacy of ROCK inhibitors according to *ROCK* polymorphism.

## Figures and Tables

**Figure 1 jcm-10-01953-f001:**
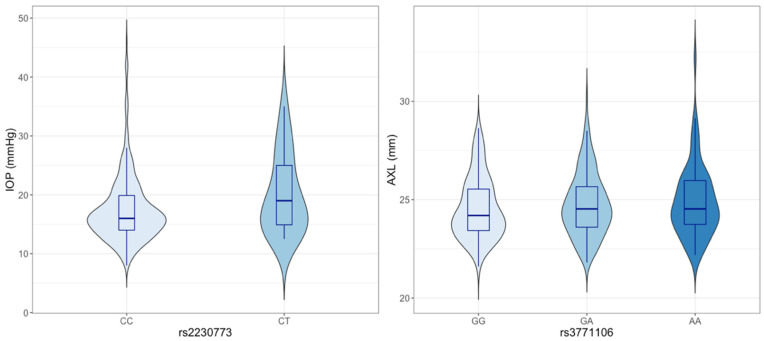
Genotype-phenotype correlations of ROCK-related variants in the entire POAG population. IOP: intraocular pressure, AXL: axial length, ROCK: Rho-associated coiled-coil kinase

**Table 1 jcm-10-01953-t001:** Target SNPs for *ROCK* polymorphism analysis

N	ID	Tagging SNP (*r*^2^ = 1)	Gene	Chr	Position	Location	AA Change
1	rs2271621	rs2290156	*ROCK2*	2	11193868	intron	
2	rs34945852		*ROCK2*	2	11197557	missense	K997M
3	rs202027620		*ROCK2*	2	11200967	missense	T881I
4	rs35768389		*ROCK2*	2	11214974	missense	D515V
5	rs190769228		*ROCK2*	2	11215516	missense	N445H
6	rs2230773		*ROCK2*	2	11249688	synonymous	
7	rs3771106	rs2230774, rs1515219, rs726843, rs978906	*ROCK2*	2	11249986	intron	
8	rs965665		*ROCK2*	2	11258749	intron	
9	rs10178332		*ROCK2*	2	11268891	intron	
10	rs1130157		*ROCK2*	2	11286615	5’UTR	
11	rs6755196		*ROCK2*	2	11320199	intron	
12	rs10929732		*ROCK2*	2	11343366	intron	
13	rs75122528		*ROCK1*	18	20947821	3’UTR	
14	rs2847081		*ROCK1*	18	20948500	3’UTR	
15	rs8089184		*ROCK1*	18	20970650	intron	
16	rs288980		*ROCK1*	18	21029619	intron	
17	rs288979	rs7239317	*ROCK1*	18	21031282	intron	
18	rs2127958		*ROCK1*	18	21073649	intron	
19	rs1481280		*ROCK1*	18	21075490	intron	
20	rs1006881	rs11874761	*ROCK1*	18	21101332	intron	
21	rs35996865		*ROCK1*	18	21112383	5’near	
22	rs10083915		*ROCK1*	18	21120994	5’near	
23	rs11873284		*ROCK1*	18	21135030	5’near	
24	rs1515210		*ROCK1*	18	29283684	5’	

SNP: single nucleotide polymorphism, ROCK: rho-kinase, Chr: chromosome, AA: amino acid, UTR: untranslated region.

**Table 2 jcm-10-01953-t002:** Subject demographics

**Variable**	**POAG (*n* = 363)**		**Healthy (*n* = 213)**	***p*** **-Value**
Age, year	54.0 ± 13.7		54.6 ± 9.7	0.55 *
NTG	54.3 ± 13.3		54.6 ± 9.7	0.76 *
HTG	53.1 ± 15.3		54.6 ± 9.7	0.40 *
Sex (Female), *n* (%)	180 (49.6%)		93 (43.7%)	0.17 †
NTG	155 (55.0%)		**93 (43.7%)**	**0.016** †
HTG	25 (30.9%)		93 (43.7%)	0.06 †
	**NTG (*n* = 282)**	**HTG (*n* = 81)**		
Baseline IOP, mm Hg	**15.3 ± 3.0**	**26.0 ± 6.5**		**<0.001** *
CCT, µm	**532.9 ± 35.9**	**546.9 ± 30.2**		**0.001** *
AXL, mm	24.7 ± 1.6	24.7 ± 1.8		0.96 *
DH history	**36 (12.8%)**	**3 (3.7%)**		**0.034** ^†^
Rim area, mm^2^	**0.77 ± 0.20**	**0.67 ± 0.29**		**0.013** *
Disc area, mm^2^	1.95 ± 0.50	1.93 ± 0.40		0.80 *
Vertical C/D	**0.77 ± 0.11**	**0.81 ± 0.11**		**0.024** *
Average RNFL thickness, μM	**70.9 ± 11.6**	**64.3 ± 14.3**		**0.001** *
Average GCIPL thickness, μM	**67.9 ± 8.8**	**63.5 ± 11.0**		**0.004** *
MD of VF, dB	−**8.4 ± 6.4**	−**14.5 ± 10.0**		**<0.001** *

Data are presented as mean ± standard deviation values. Statistically significant values are shown in bold. * Comparison performed using Student’s *t*-test, ^†^ Comparison performed using chi-square test. POAG: primary open-angle glaucoma, NTG: normal-tension glaucoma, HTG: high-tension glaucoma. IOP: intraocular pressure, CCT: central corneal thickness, AXL: axial length, DH: disc hemorrhage, C/D: cup-to-disc ratio, RNFL: retinal nerve fiber layer, GCIPL: ganglion cell-inner plexiform layer, MD: mean deviation, VF: visual field.

**Table 3 jcm-10-01953-t003:** Genetic association of *ROCK* polymorphism with POAG

Gene	ID	Alleles	POAG			
			Case MAF	Control MAF	OR (95%CI)	*p*-Value
*ROCK2*	rs2271621	G > T	0.486	0.479	1.03 (0.81, 1.32)	0.80
*ROCK2*	rs202027620	G > A	0.000	0.005		
*ROCK2*	rs2230773	C > T	0.024	0.024	1.04 (0.47, 2.33)	0.92
*ROCK2*	rs3771106	G > A	0.448	0.433	1.06 (0.83, 1.36)	0.63
*ROCK2*	rs965665	G > C	0.008	0.002	3.69 (0.44, 31.01)	0.17
*ROCK2*	rs10178332	A > C	0.008	0.002	3.69 (0.44, 31.01)	0.17
*ROCK2*	rs6755196	G > A	0.008	0.002	3.68 (0.44, 30.94)	0.17
*ROCK2*	rs10929732	G > A	0.036	0.045	0.80 (0.44, 1.47)	0.48
*ROCK1*	rs75122528	A > T	0.300	0.319	0.92 (0.71, 1.19)	0.54
*ROCK1*	rs2847081	T > C	0.138	0.127	1.09 (0.76, 1.56)	0.65
*ROCK1*	rs8089184	T > C	0.145	0.131	1.11 (0.78, 1.58)	0.56
*ROCK1*	rs288980	T > C	0.475	0.474	1.00 (0.79, 1.27)	0.99
*ROCK1*	rs288979	A > G	0.138	0.129	1.08 (0.76, 1.54)	0.67
*ROCK1*	rs2127958	T > C	0.441	0.445	0.99 (0.78, 1.25)	0.90
*ROCK1*	rs1481280	C > A	0.300	0.312	0.95 (0.74, 1.23)	0.72
*ROCK1*	rs1006881	G > A	0.140	0.129	1.10 (0.77, 1.56)	0.61
*ROCK1*	rs35996865	T > G	0.138	0.129	1.08 (0.76, 1.54)	0.68
*ROCK1*	rs10083915	A > G	0.141	0.129	1.10 (0.77, 1.56)	0.60
*ROCK1*	rs11873284	A > G	0.141	0.129	1.10 (0.77, 1.57)	0.60
*ROCK1*	rs1515210	C > G	0.183	0.181	1.03 (0.75, 1.41)	0.86

POAG: primary open-angle glaucoma, *ROCK*: rho-kinase, MAF: minor allele frequency, OR: odds ratio, CI: confidence interval. SNPs rs34945852, rs35768389, rs190769228, and rs1130157 in *ROCK2* were monomorphic in cases and controls and were removed from the table.

**Table 4 jcm-10-01953-t004:** Genetic association of *ROCK* polymorphism with NTG and HTG

Gene	ID	Alleles	Control MAF	NTG			HTG		
				Case MAF	OR (95%CI)	*p*-Value	Case MAF	OR (95%CI)	*p*-Value
*ROCK2*	rs2271621	G > T	0.479	0.491	1.05 (0.81, 1.36)	0.71	0.468	0.96 (0.66, 1.42)	0.85
*ROCK2*	rs202027620	G > A	0.005	0.000			0.000		
*ROCK2*	rs2230773	C > T	0.024	0.020	0.83 (0.34, 2.00)	0.67	0.039	1.59 (0.55, 4.56)	0.40
*ROCK2*	rs3771106	G > A	0.433	0.453	1.09 (0.84, 1.41)	0.54	0.429	1.00 (0.67, 1.47)	0.99
*ROCK2*	rs965665	G > C	0.002	0.011	4.94 (0.58, 41.83)	0.09	0.000		
*ROCK2*	rs10178332	A > C	0.002	0.011	4.94 (0.58, 41.83)	0.09	0.000		
*ROCK2*	rs6755196	G > A	0.002	0.011	4.92 (0.58, 41.72)	0.09	0.000		
*ROCK2*	rs10929732	G > A	0.045	0.036	0.78 (0.41, 1.49)	0.45	0.039	0.83 (0.33, 2.09)	0.69
*ROCK1*	rs75122528	A > T	0.319	0.282	0.86 (0.65, 1.13)	0.27	0.364	1.19 (0.82, 1.74)	0.36
*ROCK1*	rs2847081	T > C	0.127	0.153	1.22 (0.84, 1.77)	0.28	0.076	0.56 (0.28, 1.14)	0.09
*ROCK1*	rs8089184	T > C	0.131	0.164	1.28 (0.89, 1.84)	0.18	0.078	0.56 (0.29, 1.08)	0.07
*ROCK1*	rs288980	T > C	0.474	0.475	1.00 (0.78, 1.29)	1.00	0.474	0.99 (0.69, 1.42)	0.97
*ROCK1*	rs288979	A > G	0.129	0.157	1.26 (0.87, 1.82)	0.22	**0.071**	**0.52 (0.26, 1.03)**	**0.045**
*ROCK1*	rs2127958	T > C	0.445	0.441	0.99 (0.77, 1.28)	0.95	0.442	0.98 (0.68, 1.40)	0.90
*ROCK1*	rs1481280	C > A	0.312	0.280	0.88 (0.67, 1.15)	0.34	0.370	1.25 (0.87, 1.82)	0.23
*ROCK1*	**rs1006881**	**G > A**	0.129	0.159	1.28 (0.89, 1.84)	0.19	**0.071**	**0.52 (0.26, 1.03)**	**0.045**
*ROCK1*	**rs35996865**	**T > G**	0.129	0.156	1.26 (0.87, 1.82)	0.22	**0.071**	**0.52 (0.26, 1.03)**	**0.045**
*ROCK1*	**rs10083915**	**A > G**	0.129	0.161	1.28 (0.89, 1.83)	0.19	**0.071**	**0.52 (0.26, 1.03)**	**0.045**
*ROCK1*	**rs11873284**	**A > G**	0.129	0.160	1.28 (0.89, 1.85)	0.18	**0.071**	**0.52 (0.26, 1.03)**	**0.045**
*ROCK1*	rs1515210	C > G	0.181	0.185	1.05 (0.75, 1.46)	0.78	0.175	0.94 (0.58, 1.52)	0.80

Statistically significant values are shown in bold. NTG: normal-tension glaucoma, HTG: high-tension glaucoma, *ROCK*: rho-kinase, MAF: minor allele frequency, OR: odds ratio, CI: confidence interval. SNPs rs34945852, rs35768389, rs190769228, and rs1130157 in *ROCK2* were monomorphic in cases and controls and were removed from the table.

## Data Availability

The data presented in this study are available on request from the corresponding author. The data are not publicly available due to patient privacy issues.

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
