# Peer review of "Genetic Risk and Phenotype Correlation of Primary Open-Angle Glaucoma Based on Rho-Kinase Gene Polymorphisms"

_jcm, 2021, doi:10.3390/jcm10091953_

Round 1
Reviewer 1 Report
The Rho/ROCK pathway is mainly involved in the regulation of IOP, and therefore is a potential target for new drugs in Glaucoma care. The possibility also exist that alterations of the Rho kinase gene/protein may affect the control and the regulation of the IOP, and be associated with IOP elevation. Indeed, the authors described a feeble association of some SNPs with the risk of HTG, and not with NTG, where IOP might be less important, and blood flow more relevant. The association appears to be not linked to the cause of elevated IOP, but only a collateral finding, not better defined in terms of the path that leads from the mutation to IOP dysregulation. In order to give more meaning to their findings, it might be worthwhile to consider the POAG population from a different point of view: those responsive and those unresponsive to IOP regulation by drugs inhibiting the Rho kinase pathway, and see whether some of the SNPs might influence the efficacy of such drugs.
Reviewer 2 Report
The article entitled “Genetic Risk and Phenotype Correlation of Primary Open-Angle Glaucoma Based on Rho-kinase Gene Polymorphisms” investigates the role of Rho-associated coiled-coil kinase gene polymorphism on the risks of primary open-angle glaucoma (POAG) and the possible correlation of this polymorphism with the POAG phenotype in the Korean population. The article appears overall clear. However, although the content is interesting, it lacks novelty, since a previously published paper (2016) examined the same gene polymorphisms, in a Turkish population.
- Investigation of the Rho-kinase Gene Polymorphism in Primary Open-angle Glaucoma; DOI: 10.3109/13816810.2014.895016;
Furthermore, there are specific points that should be clarified:
- Methods
- Line 104: Please add a table with the primer’s sequences.
- Data analysis
In this section is not reported the student t-test statistical analysis, although it is mentioned in table 2. Furthermore, it is not also reported the software and the relative version used to perform the statistical analysis. Please add these details.
Results
Line 128. The values related to sex are expressed as a percentage whereas those related to age are expressed as a number. Please align them. Moreover, it is not specified if the values are expressed as ±SD or SEM. Please give this information. It is also not clear what kind of statistical analysis was performed to obtain the p values reported (t-test or chi-squared?). Please add this information in table footnotes.
In table 2 the DH history is very different between the two analyzed groups. Please add a comment on this data in the results section.
Discussion
The references reported are scarce. Line 182 please add some comments and relative references regarding the role of ROCK inhibitors.
Line 183 Please remove the italic in the reference number.
Reviewer 3 Report
Rho-associated coiled-coil kinases (ROCKs) are involved in the regulation of trabecular meshwork functions and it has been suggested that the genes are regulating intraocular pressure and RGC protection in previous studies. A previous study on single nucleotide polymorphisms (SNPs) in ROCK genes in Turkish population of primary open angle glaucoma (POAG) could not find any association. In Addition, recent genome wide studies on POAG patients also did not include the ROCKs in the list of genes associated with POAG. In this manuscript, the authors investigated associations of SNPs in ROCKs genes with open angle glaucoma with high tension (HTG) and normal tension (NTG) separately. It is interesting that the authors found a significant association of some SNPs on ROCK1 gene with only HTG, and SNPs on ROCK2 with intraocular pressure (IOP) and axial length in this study. The results in this manuscript may suggest the ROCK genes are involved in the IOP increase in glaucoma patients by regulating functions of trabecular meshwork.
The number of patients used in this study was not large enough to conclude their finding as the authors wrote in Discussion although it was enough to detect significant association. However, the study was well conducted, and the study methods are authentic and straightforward. The data are organized and presented in appropriate manners. Their results look reliable indicating the association of SNPs on ROCKs genes with HTG and IOP. I hope this study continues to accumulate more data to confirm their findings in future.
There are a few minor questions shown below.
1) In Figure1, it is not shown if the data represent all the subjects including all HTG, NTG and Control, or only NTG patients. Please describe about it. What was the reason that any minor homozygote was not found for rs2230773? What are the black dots on the violin graph?
2) There are some SNPs with zero minor allele frequency in the tables. I don't think there is any meaning to show them on the tables, although it may be good to mention about them in the text. Those SNPs can be removed them from the tables.
Round 2
Reviewer 2 Report
The revised version of the manuscript is suitable for publication.